# Differences in Puberty of Girls before and during the COVID-19 Pandemic

**DOI:** 10.3390/ijerph19084733

**Published:** 2022-04-14

**Authors:** Clariano Pires de Oliveira Neto, Rossana Santiago de Sousa Azulay, Ana Gregória Ferreira Pereira de Almeida, Maria da Glória Rodrigues Tavares, Luciana Helena Gama Vaz, Ianik Rafaela Lima Leal, Monica Elinor Alves Gama, Marizélia Rodrigues Costa Ribeiro, Gilvan Cortês Nascimento, Marcelo Magalhães, Wellyandra Costa dos Santos, Alexandre Nogueira Facundo, Manuel dos Santos Faria, Débora Cristina Ferreira Lago

**Affiliations:** 1Service of Endocrinology, University Hospital, Federal University of Maranhão, São Luis 65020-070, Brazil; rossanaendocrino@gmail.com (R.S.d.S.A.); agfpalmeida@gmail.com (A.G.F.P.d.A.); madagloria@gmail.com (M.d.G.R.T.); gilvancortes@uol.com.br (G.C.N.); alexandrenfacundo@hotmail.com (A.N.F.); 2Research Group in Clinical and Molecular Endocrinology and Metabology (ENDOCLIM), São Luis 65020-070, Brazil; magalhaes_ms@yahoo.com.br (M.M.); wellyandra.santos@discente.ufma.br (W.C.d.S.); mfaria1949@gmail.com (M.d.S.F.); deby19lago@gmail.com (D.C.F.L.); 3Postgraduate Program in Health Sciences, Federal University of Maranhão, São Luis 65080-805, Brazil; 4Service of Pediatric Endocrinology, University Hospital, Federal University of Maranhão, São Luis 65020-070, Brazil; luagamavaz@terra.com.br (L.H.G.V.); ianikleal@yahoo.com.br (I.R.L.L.); 5Department of Medicine III, Federal University of Maranhão, São Luis 65020-240, Brazil; monica.gama@ufma.br (M.E.A.G.); marizelia13@gmail.com (M.R.C.R.); 6Clinical Research Center, University Hospital, Federal University of Maranhão, São Luis 65020-070, Brazil

**Keywords:** precocious puberty, COVID-19, pandemic

## Abstract

In the COVID-19 pandemic, there was an increase in consultations for precocious puberty. We aim to analyze differences in female puberty before and during the COVID-19 pandemic. A cross-sectional analytical study was designed at the Pediatric Endocrinology Clinic of the University Hospital of the Federal University of Maranhão in São Luis, Brazil. We included 55 girls with precocious puberty, 22 who started puberty during the pandemic and 33 who started puberty before the pandemic. Clinical, anthropometric, laboratory and imaging variables were compared between groups. Statistics were performed to determine if there was a statistical difference between the groups. Girls with puberty during the pandemic had higher Z-scores for weight (1.08 ± 1.29 versus 0.69 ± 0.83; *p* = 0.04), lower ovarian volume (1.88 ± 0.95 versus 3.15 ± 2.31; *p* = 0.01), and smaller differences between thelarche noticed by the parents and the diagnosis (6.63 ± 5.21 versus 12.15 ± 9.96; *p* = 0.02). The association between precocious puberty during the pandemic with higher Z-scores for weight, lower ovarian volume, and a reduction in the time between the perception of pubertal findings by parents and the diagnosis suggests the influence of the pandemic on the normal time of puberty.

## 1. Introduction

Puberty is an essential and complex process, with a wide physiological variation of maturation, which consists of a period of transition between the immature to mature sexual state and culminates in the reproductive capacity and attainment of final stature after rapid somatic growth [1,2]. The onset and progression of this process may vary according to genetic, metabolic, socioeconomic, ethnic, and geographic characteristics, and it results in the reactivation of the hypothalamic–pituitary–gonadal (HPG) axis [3].

Pubertal onset is the result of a complex neuroendocrine system that leads to an increase in the release of gonadotropin-releasing hormone (GnRH) by the hypothalamus, with an increase in the secretion and release of gonadotropins (LH = luteinizing hormone and FSH = follicle-stimulating hormone) by the adeno -pituitary [4]. Environmental factors such as nutritional changes, obesity, and exposure to endocrine disruptors are thought to be “triggers” for puberty onset [5]. This complex endocrine alteration influenced by multiple and integrated peripheral and central signals is responsible for the development of sexual characteristics, pubertal spurt, and achievement of reproductive competence [6].

Precocious puberty in girls is defined as the development of secondary sexual characteristics before the age of 8 years due to premature activation of the HPG axis [7]. It is a rare condition, with an estimated incidence of 1:5000 to 1:10,000, more frequent in females, with a ratio of 3–23 girls: 1 boy [8,9]. In girls, it has been reported that the age and time of puberty progression have decreased worldwide, indicating a reduction in the ages of menarche and thelarche [10]. The factors related to the early onset of puberty have been increasingly understood, especially with the discovery of genetic mutations related to the release of the HPG axis, such as mutations by deletion in the MKRN3 gene of autosomal dominant inheritance and paternal transmission, which alter the release of GnRH [11].

There are several differential diagnoses in precocious puberty. Didactically, precocious puberty can be divided into central precocious puberty, peripheral precocious puberty, and variants of normality [12]. Central precocious puberty is most often idiopathic but may be due to mutations in genes involved in the modulation of GnRH secretion and central nervous system disease such tumors, infections, and trauma [4]. Peripheral precocious puberty is caused by extra pituitary secretion of gonadotropins or sex steroids, which may be of adrenal origin (congenital adrenal hyperplasia or adrenal tumors) or ovarian (autonomic follicular cysts or ovarian tumors) [13]. When pubertal events are incomplete, it is associated with normal variants of puberty (isolated early thelarche, early adrenarche, constitutional acceleration of growth and puberty) [1].

Although the most important parameter for the diagnosis of precocious puberty is the elevation of the LH value (baseline or after stimulation with GnRH analogues), it is described that pelvic ultrasound is a useful component in the differentiation with precocious thelarche, in which the consensus reports cutoff values between 1.0 and 3.0 mL of ovarian volume as a diagnostic criterion for precocious puberty [14].

In late 2019, the novel coronavirus SARS-CoV-2 (Severe Acute Respiratory Syndrome Coronavirus 2) was identified as the cause of a cluster of atypical pneumonia cases in Wuhan, China [15]. This emerging disease called COVID-19 (Coronavirus Disease 2019) has rapidly expanded, leading to a global pandemic [16]. In March 2020, in view of the high transmission rate and severity of the clinical condition, with maximum occupancy of hospital beds, restrictive measures were adopted in several cities in Brazil, leading to the closure of schools, restriction of exit from households, and introduction of the remote work.

During the lockdown and immediately after the end of the most restrictive measures, an increase in the number of outpatient consultations for suspected precocious puberty was observed in several centers worldwide [17]. In our center, we have seen a 50% increase in consultations for suspicious early puberty during the COVID-19 pandemic compared to data before the pandemic. Thus, this study was developed with the objective of evaluating pubertal development and factors related to its precocity during the period of the COVID-19 pandemic.

## 2. Materials and Methods

A cross-sectional study was carried out with a convenience sample consisting of 22 girls diagnosed with precocious puberty during the COVID-19 pandemic (July 2020 to June 2021) and followed up at the Pediatric Endocrinology outpatient Clinic of the University Hospital of the Federal University of Maranhão (HUUFMA) Maternal and Child Unit. For comparison, a control group was composed, formed by 33 girls who followed in the same service from March 2019 to February 2020 and were diagnosed with precocious puberty prior to the COVID-19 pandemic.

Patients diagnosed with precocious puberty associated with hypothalamic–pituitary malformations, neurological, neurosurgical and/or genetic diseases, psychomotor retardation, oncological diseases, or other endocrine diseases that require hormonal treatment or that may interfere with pubertal development were excluded from the study.

Precocious puberty was defined as the development of pubertal changes at age less than 8 years in girls in the presence of LH > 0.3 IU/L, LH peak > 5 IU/L on GnRH stimulation test, or ovarian volume > 2 cm^3^ at pelvic ultrasound [18].

The consultation of medical records consisted of accessing information on physical examination and complementary exams. Data on the diagnosis of height, weight, body mass index (BMI), and pubertal stage (Tanner scale) were obtained. The growth velocity (GV) in centimeters per years (cm/year) was calculated, which was obtained after a follow-up interval of at least 6 months. BMI was calculated by dividing weight in kilograms (kg) by the square of height in meters (m^2^). Height, weight and BMI were normalized for chronological age by conversion to standard deviation (SD) scores according to World Health Organization (WHO) graphs [19].

Pubertal development was classified according to the criteria of Marshall and Tanner [20]. Age at puberty onset was defined as the age at Tanner stage M2 (thelarche), perceived by parents, and confirmed in medical evaluation with the aid of laboratory tests, pelvic ultrasound, and radiography of hands for bone age (BA). Laboratory evaluation consisted of LH, FSH, and estradiol dosages. Skeletal maturation was expressed as BA, and it was calculated according to the Greulich and Pyle method [21] and as the difference between BA and chronological age (BA–CA).

The data obtained were tabulated in an Excel spreadsheet. Categorical variables were described in tables containing absolute and relative frequencies and quantitative variables with mean ± standard deviation. Normality was assessed using the Shapiro–Wilk test. Categorical variables were evaluated using the chi-square or Fisher’s exact tests, depending on the absolute frequency of the groups. For continuous variables, the *t*-test was used for variables with normal distribution and the Mann–Whitney test was used for non-parametric variables. All analyses were performed using the Statistical Data Analysis software Version 15.

This study was approved by the Research Ethics Committee of the University Hospital of the Federal University of Maranhão under protocol number 4922948. Secondary data from medical records, recorded by the attending physicians during medical consultations, were used. The data were tabulated without identifying the research subjects in the other stages. Waiver on the informed consent was requested, via specific form, according to item V of article 1 of Resolution 510/2016 of the National Health Commission of the Ministry of Health of Brazil, which deals with the use of secondary data. The justification for the exemption was based on the circumstances of the pandemic and the need to restrict the movement of people and also because some children reside in other municipalities, making difficult to travel and attend to sign the consent form.

## 3. Results

Clinical data and laboratory results were analyzed in 55 girls. Twenty-two patients were diagnosed with precocious puberty after the onset of the COVID-19 pandemic. Clinical, anthropometric, laboratory, and radiological data from these patients were compared with data from 33 patients followed by precocious puberty in the year immediately prior to the onset of the pandemic (March 2019 to February 2020).

In comparison with the control group, the girls who developed precocious puberty during the pandemic had a more advanced age at diagnosis (7.15 versus 6.74 years; *p* = 0.10), but time from the parents’ perception of the breast to diagnosis was lower (6.65 versus 12.15 months; *p* = 0.02). They were taller, with a mean Z-score for height of 0.76 versus 0.22 (*p* = 0.07) and slightly larger GV (9.3 versus 8.59 cm/year; *p* = 0.14). Weight was higher in girls diagnosed during the pandemic with a mean Z-score of 1.08 versus 0.69 (*p* = 0.04) and Z-score for BMI 0.97 versus 0.75 (*p* = 0.07). Obesity was more prevalent in the group that developed puberty during the pandemic (36.4% versus 18.2%), but without statistical significance, as was Tanner’s staging at diagnosis (*p* = 0.16) (Table 1).

BA at diagnosis was similar between the groups (9.55 versus 9.82 years; *p* = 0.19) as was the BA–CA difference (1.96 versus 1.87 years; *p* = 0.36). Regarding the laboratory evaluation, there was no statistical difference between the groups for the dosages of LH, FSH and estradiol. Ovarian volume was significantly higher in girls diagnosed before the pandemic (3.15 mL) compared to those diagnosed during the pandemic (1.88 mL) with *p* = 0.01 (Table 2) (Figure 1).

## 4. Discussion

Our study is the first in Brazil to study puberty in girls during the COVID-19 pandemic. In our results, we found that pubertal anticipation during the pandemic was associated with higher Z-score for weight, lower ovarian volume, and shorter time between parents’ perception of thelarche and diagnosis. In our research in the literature, only one study investigated the impact of the pandemic on the pubertal development of girls, with results different from ours, as they found a statistically significant difference for the variables age of M2, age at diagnosis, Tanner staging, LH and FSH [6].

We observed a reduction of half in the time between perception of parents and diagnosis of M2 stage comparing pre-pandemic and during the pandemic data (12 months versus 6 months). Parents’ perception of their children’s pubertal development is generally seen as satisfactory, but there is a tendency to underestimate pubertal staging [22]. Two main possibilities can be raised: the time in isolation made it possible to better observe the bodily changes of puberty, or it presented itself in a more accelerated way. Such clarifications will only be possible with the passage of time, according to the pubertal progression pace of the evaluated girls.

In girls, idiopathic precocious puberty accounts for about 90% of all cases, although advances in genetics and diagnostic imaging make it possible to review these statistics [4]. In the context of the COVID-19 pandemic, environmental factors gain relevance as potential triggers of puberty anticipation: SARS-CoV-2 infection, weight gain, excessive use of electronic devices and psychological triggers [23].

Childhood obesity and changes in children’s eating patterns have been related to the trend of puberty anticipation, especially in girls, observed over the last few decades [24]. In malnourished children, rapid weight gain is a factor related to the onset of pubertal events [25]. A meta-analysis of 11 cohort studies showed that obesity was an important risk factor for precocious puberty in girls [26]. In our results, we found a higher prevalence of obesity among girls who started puberty during the pandemic (36.36% against 18.18%), but without statistical relevance. A higher mean Z-score for BMI was also observed among girls with puberty during the pandemic, with a non-significant difference. However, the difference in the variable Z-score for weight was considered statistically significant between the groups (*p* = 0.04). This finding may be based on the increase in calorie intake, changes in dietary patterns and reduction in physical activity during the pandemic, however with a modest effect on BMI, which was probably due to the greater height of the pubertal period, which is not enough to explain the increase in the frequency of precocious puberty as highlighted in a recent review [23]. A hypothesis that arises in light of these findings concerns the tendency of preschool children’s body compositions to change over the years, with an increase in adipose tissue and a reduction in muscle mass, which is a process known as latent obesity [27], which would be a probable explanation for the small difference in BMI found between the groups.

During the pandemic, especially in the period of isolation, there was a significant change in the dietary pattern of children, especially with an increase in the intake of bread and sweets [28]. In the Brazilian population, there was an increase in the consumption of simple carbohydrates and sugary drinks associated with a reduction in the consumption of fruits, juices and vegetables [29]. Allied to this, there was a drastic decrease in the levels of physical activity, mainly walking and cycling, due to issues intrinsic to the pandemic and a reduction in the movement of people, which contributed to a general increase in BMI and the frequency of obesity [30,31]. In our study, this situation was suggested by a statistically significant difference in the variable Z-score for weight between the pre-pandemic and during the pandemic groups.

Increase in adipose tissue is capable of promoting an increase in leptin concentrations and a reduction in ghrelin concentrations [32]. These two peripheral mediators have been increasingly studied in the context of puberty with evidence that leptin promotes pulsatile GnRH secretion and ghrelin suppresses the pulsatility of GnRH secretion [33,34]. A recently published study identified that girls with onset of puberty during the pandemic had significantly lower levels of ghrelin when compared to a control group, with a mechanism probably related to the reduction in serum concentration of MKRN3, with loss of negative regulation on suppressing the pulsatile secretion of GnHR and promoting the onset of puberty [35].

One finding from our study that arouses curiosity is that the ovarian volume of girls during the pandemic is smaller than in girls with puberty before the pandemic (*p* = 0.01). Girls with early thelarche have higher Z-scores for weight and higher percentage of body fat than age-matched girls without thelarche [36]. Early thelarche in obese girls is not a result of GnRH activation but an isolated consequence of increased aromatase activity in adipose tissue, where androgens are converted to estrogens [37]. Thus, it is likely that the girls who developed puberty during the pandemic in our study had smaller ovarian volume due to the development of pubertal milestones secondary to the production of estrogens in adipose tissue and not by the ovaries.

Other factors that we were unable to assess in this study may also be related to puberty anticipation in the period during the pandemic. A direct effect of COVID 19 infection should be considered, given the common embryological origin between the olfactory bulb (affected in the early stages of SARS-CoV-2 infection) and the GnRH-producing hypothalamic neurons [38,39]. There is also estimated a greater contact with endocrine disruptors (exogenous compounds capable of binding to hormone receptors) from household objects during the pandemic period due to the longer isolation period [40]. They are used by the industry in various products such as plastics, solvents, lubricants, pesticides, and additives [41] or produced by nature, such as phytoestrogens, presents in nuts, soy products, cereals and breads, interfering with the endogenous endocrine function [42]. The abundance of endocrine disruptors associated with the secular trend of pubertal anticipation has led researchers to associate them with precocious puberty, especially those with estrogenic activity [43].

Excessive use of electronic devices, whether for educational or recreational activities, may have contributed to physical inactivity and increased incidence of obesity, in addition to promoting changes in melanin levels, related to the onset of pubertal events [6,44]. A recent Italian study found that girls who started precocious puberty during the pandemic had longer use of electronic devices and lower levels of melatonin [45]. It is possible that increased screen tie with reduced melatonin levels triggers endocrine changes that lead to the onset of pubertal development at an earlier age. Finally, psychological factors can also be important in pubertal development during the pandemic, with an impact on children’s mental health and well-being, leading to stressful situations that can be “triggers” for puberty onset [28].

The strengths of this study are: (1) it was carried out in a reference center for the population of Maranhão; (2) it is pioneering, as no studies were found on this subject in the Brazilian population; (3) it describes weight as a trigger for puberty anticipation, as in Italian studies [6,17]. This study has important limitations, among which we mention: (1) the small sample size, given the still short period of time during the pandemic; (2) the non-homogeneous control group, due to the inclusion of girls in different pubertal stages; (3) follow-up by different evaluators during the pandemic, due to the need to withdraw from face-to-face activities in the context of the pandemic; (4) the use of self-reported information by parents, which may not be accurate; and (5) lack of data on infection status of COVID-19, calorie consumption, physical activity, screen time, percentage of adipose tissue and exposure to endocrine disruptors.

## 5. Conclusions

We found an association between precocious puberty during the pandemic with higher Z-score for weight and lower ovarian volume. These findings are probably related to the changes observed during the pandemic, such as the increase in caloric food intake, physical inactivity and increased use of electronic devices, allowing an increase in adiposity and estrogen production in adipose tissue in an amount to promote puberty onset without increasing the ovarian volume.

In addition, we noticed a shorter interval between the perception of thelarche by the parents and the diagnosis of precocious puberty, which was possibly due to the greater observation of pubertal findings by the parents of the children or due to a faster rate of puberty progression.

According to other authors, we suggest an adoption of health lifestyle habits especially during periods of social isolation, with a varied and balanced diet and regular practice of physical activity. It is also essential to reduce the time of use of electronic devices, although many school activities use this as an alternative teaching method. Parents need to pay attention to the body changes in puberty in their children, reinforcing the need for medical consultation if any abnormality is suspected. Many changes and causal relationship with the pandemic period can only be better clarified in the future.

This study is one of the few that evaluated precocious puberty in the context of the COVID-19 pandemic. More studies are needed to correlate the increase in precocious puberty observed in girls to specific pathogenic factors in the context of the pandemic.

## Figures and Tables

**Figure 1 ijerph-19-04733-f001:**
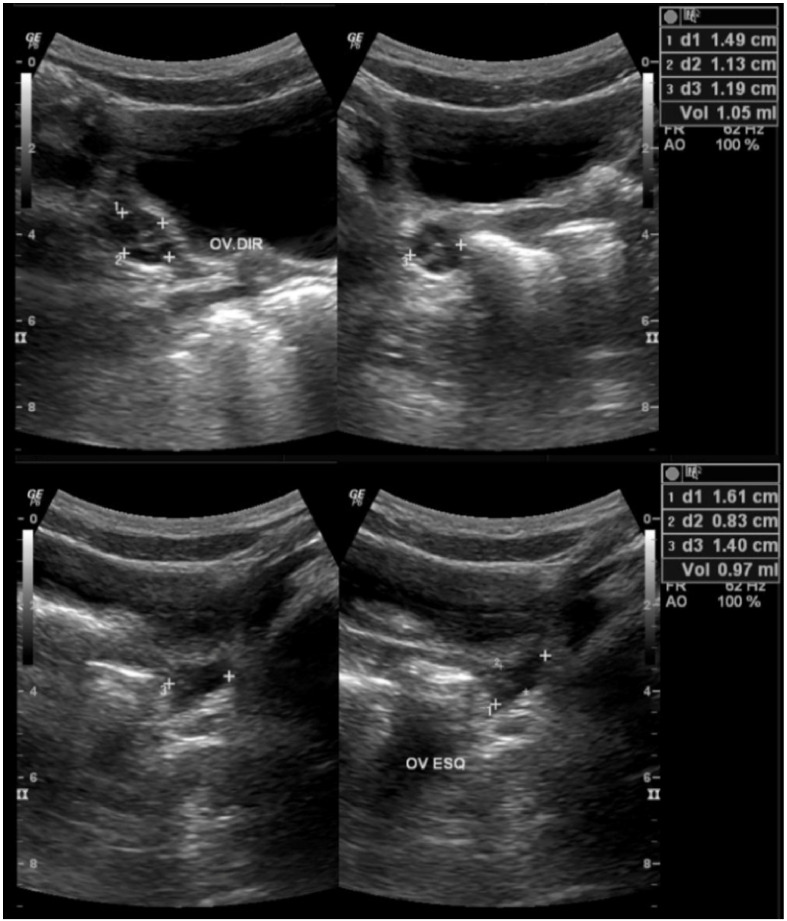
Ovarian ultrasound of a girl diagnosed with precocious puberty during the COVID-19 pandemic showing a reduced pubertal ovary. The patient started puberty at 7 years and 3 months (May 2020); August 2020: height 1.67 SDS, weight 1.53 SDS, BMI 1.1 SDS, Tanner stage 3.

**Table 1 ijerph-19-04733-t001:** Clinical and anthropometric characteristics of girls with precocious puberty before and during the COVID-19 pandemic (*n* = 55).

Variable	Pre-Pandemic *n* = 33	During Pandemic *n* = 22	*p*-Value
M2 age (in years)			
Mean ± SD	6.74 ± 0.85	7.15 ± 0.71	0.10 T
Age at diagnosis (in years)			
Mean ± SD	7.86 ± 0.82	7.70 ± 0.62	0.22 T
Time from M2 to diagnosis (in months)			
Mean ± SD	12.15 ± 9.96	6.63 ± 5.21	0.02 M
Z-score to height			
Mean ± SD	0.22 ± 0.98	0.76 ± 1.19	0.07 M
GV (in cm/year)			
Mean ± SD	8.59 ± 2.30	9.35 ± 2.93	0.14 T
Z-score to weight			
Mean ± SD	0.69 ± 0.83	1.08 ± 1.29	0.04 M
Z-score to BMI			
Mean ± SD	0.75 ± 0.86	0.96 ± 1.34	0.07 M
Normal	21 (63.64%)	9 (40.91%)	0.20 Q
Overweight	6 (18.18%)	5 (22.73%)	
Obesity	6 (18.18%)	8 (36.36%)	
Tanner Stage			0.16 E
2	12 (36.36%)	7 (31.82%)	
3	18 (54.56%)	15 (68.18%)	
4	3 (9.09%)	0	

M2—Telarche; GV—Growth velocity; SD—Standard deviation; T—Test for independent samples; M—Mann–Whitney; Q—Chi-square; E—Fisher’s exact.

**Table 2 ijerph-19-04733-t002:** Laboratory and radiological characteristics of girls with precocious puberty before and during the COVID-19 pandemic (*n* = 55).

Variable	Pre Pandemic *n* = 33	During Pandemic *n* = 22	*p*-Value
BA (in years)			
Mean ± SD	9.82 ± 1.19	9.55 ± 1.34	0.19 M
BA–CA (in years)			
Mean ± SD	1.96 ± 0.87	1.87 ± 1.00	0.36 T
LH (IU/L)			
Mean ± SD	1.35 ± 1.72	1.28 ± 2.17	0.45 M
FSH (IU/L)			
Mean ± SD	4.11 ± 2.64	4.33 ± 3.05	0.36 M
Estradiol (pmol/L)			
Mean ± SD	27.44 ± 18.80	25.61 ± 15.97	0.40 M
Ovarian Volume (mL)			
Mean ± SD	3.15 ± 2.31	1.88 ± 0.95	0.01 M

BA—Bone age; CA—Chronological age; SD—Standard deviation; LH—Luteinizing hormone; FSH—Follicle-stimulating hormone; T—T test for independent samples; M—Mann–Whitney.

## Data Availability

The datasets used and analyzed during the current study are available from the corresponding author upon reasonable request.

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
