# Peer review of "Differences in Puberty of Girls before and during the COVID-19 Pandemic"

_ijerph, 2022, doi:10.3390/ijerph19084733_

Round 1

Reviewer 1 Report

  1. The topic is very important, the author addressed the study with classic possible causes, authors have to discuss in detail other factors as most of the data were not significant such as very recent study in 2022 who reported that (serum concentrations of MKRN3 and ghrelin, and indicated ghrelin as a potential regulatory mechanism of puberty. https://doi.org/10.3389/fendo.2022.839895). 
  2. There some notes have been posted on the manuscript it self
  3. Since most of results are not significants but the authors clarified this honestly. It is good attitude ethically. But it shouldn't be base of the discussion. So other possible factors must to be addressed even if theoretically such as they mentioned melatonin and fat' estrogen which needs more details 
  4. Other causes for prematuration such as:  
  5. The authors didn't get ethical approval for this study (IRB committee ) of their institute (Authors have to provide no # and date of this approval). Participants or their parents approval are not sufficient to carry out the study on human or their data. 
  6. Results of ultrasound figures of ovary must be presented in this work 
  7. No 2022 references were cited in this work such as for example only https://doi.org/10.1530/EC-21-    https://doi.org/10.3389/fendo.2022.839895). 

Author Response

Response to reviewer 1

1. The topic is very important, the author addressed the study with classic possible causes, authors have to discuss in detail other factors as most of the data were not significant such as very recent study in 2022 who reported that (serum concentrations of MKRN3 and ghrelin, and indicated ghrelin as a potential regulatory mechanism of puberty. https://doi.org/10.3389/fendo.2022.839895). 

     Response: The authors are grateful for the suggestion and indication of the article and have added a paragraph (lines 215 to 223) on ghrelin and MKRN3 concentrations in precocious puberty.

     “Increase in adipose tissue is capable of promoting an increase in leptin concentrations and a reduction in ghrelin concentrations [32]. These two peripheral mediators have been increasingly studied in the context of puberty with evidence that leptin promotes pulsatile GnRH secretion and ghrelin suppresses the pulsatility of GnRH secretion [33,34]. A recently published study identified that girls with onset of puberty during the pandemic had significantly lower levels of ghrelin when compared to a control group, with a mechanism probably related to the reduction of serum concentration of MKRN3, with loss of negative regulation on suppressing the pulsatile secretion of GnHR and promoting the onset of puberty [35].

2. There some notes have been posted on the manuscript itself.

     Response: Notes have been deleted in the current version of the manuscript. Correction points were made using "Track Changes" function according to the editor instructions.

3. Since most of results are not significants but the authors clarified this honestly. It is good attitude ethically. But it shouldn't be base of the discussion. So other possible factors must to be addressed even if theoretically such as they mentioned melatonin and fat' estrogen which needs more details 

     Response: The authors detailed in the discussion the role of melatonin (lines 246 to 252) and endocrine disruptors including phytoestrogens (lines 237 to 245) in precocious puberty.

    “Excessive use of electronic devices, whether for educational or recreational activities, may have contributed to physical inactivity and increased incidence of obesity, in addition to promoting changes in melanin levels, related to the onset of pubertal events [6,44]. Recent Italian study found that girls who started precocious puberty during the pandemic had longer use of electronic devices and lower levels of melatonin [45]. It is possible that increased screen tie with reduced melatonin levels triggers endocrine changes that lead to the onset of pubertal development at an earlier age.“

     “It is also estimated a greater contact with endocrine disruptors (exogenous compounds capable of binding to hormone receptors) from household objects during the pandemic period due to the longer isolation period [40]. They are used by the industry in various products such as plastics, solvents, lubricants, pesticides and additives [41] or produced by nature, such as phytoestrogens, presents in nuts, soy products, cereals and breads, interfering with the endogenous endocrine function [42]. The abundance of endocrine disruptors associated with the secular trend of pubertal anticipation has led researchers to associate them with precocious puberty, especially those with estrogenic activity [43].”

4. Other causes for prematuration such as:  

     Response: A paragraph indicating other causes of precocious puberty has been included in the introduction (lines 54 to 63).

     “There are several differential diagnoses in precocious puberty. Didactically, precocious puberty can be divided into central precocious puberty, peripheral precocious puberty and variants of normality [12]. Central precocious puberty is most often idiopathic but may be due to mutations in gen involved in the modulation of GnRH secretion and central nervous system disease such tumors, infections and trauma [4]. Peripheral precocious puberty is caused by extra pituitary secretion of gonadotropins or sex steroids, which may be of adrenal origin (congenital adrenal hyperplasia or adrenal tumors) or ovarian (autonomic follicular cysts or ovarian tumors) [13]. When pubertal events are incomplete, it is associated with normal variants of puberty (isolated early thelarche, early adrenarche, constitutional acceleration of growth and puberty) [1].”

5. The authors didn't get ethical approval for this study (IRB committee ) of their institute (Authors have to provide no # and date of this approval). Participants or their parents approval are not sufficient to carry out the study on human or their data. 

     Response: The authors modified the paragraph on Ethics Committee approval and included the requested information (lines 121 to 130).

     “This study was approved by the Research Ethics Committee of the University Hospital of the Federal University of Maranhão under protocol number 4922948. Secondary data from medical records, recorded by the attending physicians during medical consultations, were used. The data were tabulated without identifying the research subjects in the other stages. Waiver on the Informed Consent was requested, via specific form, according to item V of article 1 of Resolution 510/2016 of the National Health Commission of the Ministry of Health of Brazil, which deals with the use of secondary data. The justification for the exemption was based on the circumstances of the pandemic and the need to restrict the movement of people and also because some children reside in other municipalities, making difficult to travel and attend to sign the Consent Form.

6. Results of ultrasound figures of ovary must be presented in this work 

     Response: Pelvic ultrasound images illustrating the ovaries was included (lines 163 to 165).

     “Figure 1. Ovarian ultrasound of a girl diagnosed with precocious puberty during the COVID-19 pandemic showing a reduced pubertal ovary. The patient started puberty at 7 years and 3 months (May 2020); August 2020: height 1.67 SDS, weight 1.53 SDS, BMI 1.1 SDS, Tanner stage 3.” 

7. No 2022 references were cited in this work such as for example only https://doi.org/10.1530/EC-21-  https://doi.org/10.3389/fendo.2022.839895). 

     Response: Once again the authors are grateful for the literature suggestions for inclusion in the manuscript (References 31 and 35).

     “31. Chioma, L.; Bizzarri, C.; Verzani, M.; Fava, D.; Salerno, M.; Capalbo, D.; Guzzetti, C.; Penta, L.; di Luigi, L.; di Iorgi, N.; et al. Sedentary Lifestyle and Precocious Puberty in Girls during the COVID-19 Pandemic: An Italian Experience. Endocrine Connections 2022, 11, doi:10.1530/ec-21-0650.”

     “35. Chen, Y.; Chen, J.; Tang, Y.; Zhang, Q.; Wang, Y.; Li, Q.; Li, X.; Weng, Z.; Huang, J.; Wang, X.; et al. Difference of Precocious Puberty Between Before and During the COVID-19 Pandemic: A Cross-Sectional Study Among Shanghai School-Aged Girls. Frontiers in Endocrinology 2022, 13.”

Reviewer 2 Report

The manuscript presents the first results of a study of early onset of puberty in Barazilian girls befor and after the COVID-19 pandemic lockdown. The research sample consists of 22 girls, which is compared with the control sample of 33 girls with precocious puberty befor the pandemic. The pandemic girls had a significantly higher body mass and, conversely, had a significantly smaller ovarian volume from all monitored parameters.

The manuscript provides a number of interesting suggestions and is also a benefit for clinical practice. The changes in puberty timing in girls during the COVID-19 pandemic, in terms of early onset rapid progression, are observed in a number of countries.  In the discussion, the authors give a possible explanations and suggestions for further research. All this can be applied to the anamnesis in clinical practice.

Comments:

Introduction:

The introduction states that the number of consultations for premature puberty in girls is increased during the pandemic. It would be interesting if the authors included in the characteristic of the research sample at least an indicative comparison of the incidence for a certain period of the time before and after the lockdown at their clinic.

Material and Methods:

I do not find any problems with the study design or the statistics.

Results:

The study has some limitations, but they are accurately discussed by the authors.

Discussion:

The discussion is clear and comprehensive. However, there is no explanation for the fact why there was no significant increase in BMI in the sample of girls COVID-19. I recommend to the authors look at the issue of so-called hidden, normal-weight (also latent) obesity, described in the current preschool children. This phenomenon could be a good explanation.

For clinical practice it would be beneficial and appropriate to supply:

  • Further develop practical recommendations for anamnestic and clinical examination of the patient.
  • More specific possible estrogen-like substances in households that could have the effect of endocrine disruptors. The authors only state the possibility of influence in one sentence in the discussion.

It is a pity that the percentage of adipose tissue was not monitored in the study. Thus, the relationship between obesity at early onset of puberty cannot be documented. I recommend to add to the study limits.

Line 209-210: in reference 3) indicates agreement with Italian studies. However, the citations [9, 18] correspond to a Danish and a British study.

  1. Bräuner, E. v; Busch, A.S.; Eckert-Lind, C.; Koch, T.; Hickey, M.; Juul, A. Trends in the Incidence of Central Precocious Puberty and Normal Variant Puberty Among Children in Denmark, 1998 to 2017. JAMA Network Open 2020, 3, e2015665–e2015665, doi:10.1001/jamanetworkopen.2020.15665.
  2. Marshall, W.A.; Tanner, J.M. Variations in Pattern of Pubertal Changes in Girls. Archives of disease in childhood 1969, 44, 291–303, doi:10.1136/adc.44.235.291.

Conclusion:

Without comments.

References:

Citation 20., line 292-293: different font format; please unite.

Author Response

The manuscript presents the first results of a study of early onset of puberty in Barazilian girls befor and after the COVID-19 pandemic lockdown. The research sample consists of 22 girls, which is compared with the control sample of 33 girls with precocious puberty befor the pandemic. The pandemic girls had a significantly higher body mass and, conversely, had a significantly smaller ovarian volume from all monitored parameters.

The manuscript provides a number of interesting suggestions and is also a benefit for clinical practice. The changes in puberty timing in girls during the COVID-19 pandemic, in terms of early onset rapid progression, are observed in a number of countries.  In the discussion, the authors give a possible explanations and suggestions for further research. All this can be applied to the anamnesis in clinical practice.

Comments:

Introduction:

1. The introduction states that the number of consultations for premature puberty in girls is increased during the pandemic. It would be interesting if the authors included in the characteristic of the research sample at least an indicative comparison of the incidence for a certain period of the time before and after the lockdown at their clinic.

     Response: The authors are grateful for the suggestion made and have included the requested data (lines 78 and 80).

     “In our center, we have seen a 50% increase in consultations for suspicious early puberty during the COVID-19 pandemic compared to data before pandemic.”

Discussion:

2. The discussion is clear and comprehensive. However, there is no explanation for the fact why there was no significant increase in BMI in the sample of girls COVID-19. I recommend to the authors look at the issue of so-called hidden, normal-weight (also latent) obesity, described in the current preschool children. This phenomenon could be a good explanation.

     Response: The authors agree with the suggestion made and have included the requested topic (lines 200 to 204).

     “A hypothesis that arise in light of this findings concerns the tendency of preschool children’s body compositions to change over the years, with an increase in adipose tissue and a reduction in muscle mass, a process known as latent obesity [27], which would be a probable explanation for the small difference in BMI found between the groups.

3. For clinical practice it would be beneficial and appropriate to supply:

Further develop practical recommendations for anamnestic and clinical examination of the patient.

     Response: A paragraph was included in conclusion with recommendations useful in clinical practice (lines 278 to 284)

     “According to other authors we suggest adoption of health lifestyle habits especially during periods of social isolation, with a varied and balanced diet and regular practice of physical activity. It is also essential to reduce the time of use of electronic devices, although many school activities use this as an alternative teaching method. Parents need to pay attention to the body changes on puberty in their children, reinforcing the need for medical consultation if any abnormality is suspected. Many changes and causal relationship with the pandemic period can only be better clarified in the future.”

4. More specific possible estrogen-like substances in households that could have the effect of endocrine disruptors. The authors only state the possibility of influence in one sentence in the discussion.

     Response: The authors added the requested information on estrogen-like substances in households that could have the effect of endocrine disruptors (lines 237 to 245)

     “It is also estimated a greater contact with endocrine disruptors (exogenous compounds capable of binding to hormone receptors) from household objects during the pandemic period due to the longer isolation period [40]. They are used by the industry in various products such as plastics, solvents, lubricants, pesticides and additives [41] or produced by nature, such as phytoestrogens, presents in nuts, soy products, cereals and breads, interfering with the endogenous endocrine function [42]. The abundance of endocrine disruptors associated with the secular trend of pubertal anticipation has led researchers to associate them with precocious puberty, especially those with estrogenic activity [43].

5. It is a pity that the percentage of adipose tissue was not monitored in the study. Thus, the relationship between obesity at early onset of puberty cannot be documented. I recommend to add to the study limits.

     Response: The measurement of body adiposity by adipometer, bioimpedance and bone densitometry is not routine in the endocrinology service and the authors agree that it is a limitation of the study (line 265). However, we used the BMI marker, as recommended by the WHO, capable o defining the diagnosis of overweight and obesity (lines 103 to 105).

     “5) lack of data on infection status of COVID-19, calorie consumption, physical activity, screen time, percentage of adipose tissue and exposure to endocrine disruptors.”

     “Height, weight and BMI were normalized for chronological age by conversion to standard deviation (SD) scores according to World Health Organization (WHO) graphs [19].”

6. Line 209-210: in reference 3) indicates agreement with Italian studies. However, the citations [9, 18] correspond to a Danish and British study

     Response: The references have been corrected to indicate appropriate studies (references 6 and 17)

7. Citation 20., line 292-293: different font format; please unite

     Response: The cited reference was re-edited in the journal format 

Reviewer 3 Report

Dear authors,

in order to improve the quality of your article and make it more suitable for publishing I would suggest you to:

  • modify the title (obviously you have not presented the results after the pandemic, instead of that you can choose between during or, even better, after the onset of the COVID-19 pandemic)
  • adjust the period of your research in whole manuscript (in the title and  abstract - after, in the method - during, and in the results - after the onset)
  • since you have analysed results just before and immediately after the onset of the pandemic, without possibility to discuss the influence of COVID-19 pandemic to precocious puberty of girls, I would suggest you to check if the girls from second group had positive test results for coronavirus, and to analyse the difference, if exists.
  • check the references citation (i.e. at the row 210 you have mentioned Italian studies but refered to another articles - 9,18).

Author Response

In order to improve the quality of your article and make it more suitable for publishing I would suggest you to:

1. Modify the title (obviously you have not presented the results after the pandemic, instead of that you can choose between during or, even better, after the onset of the COVID-19 pandemic)

     Response: The authors are grateful for the suggestion and have modified the title to “Differences in puberty of girls before and during the COVID-19 pandemic

2. Adjust the period of your research in whole manuscript (in the title and  abstract - after, in the method - during, and in the results - after the onset)

     Response: All suggested changes have been made (lines 2, 16, 19, 22, 25, 79, 81, 138, 142, 144, 150, 154, 157, 163, 168, 175, 192, 194, 214, 225, 230, 234, 253, 260 and 268)

3. Since you have analysed results just before and immediately after the onset of the pandemic, without possibility to discuss the influence of COVID-19 pandemic to precocious puberty of girls, I would suggest you to check if the girls from second group had positive test results for coronavirus, and to analyse the difference, if exists.

     Response: COVID-19 infection status in girls in the second group is not available. This data was put as a limitation of the study (line 264)

     “5) lack of data on infection status of COVID-19, calorie consumption, physical activity, screen time, percentage of adipose tissue and exposure to endocrine disruptors.”

4. Check the references citation (i.e. at the row 210 you have mentioned Italian studies but refered to another articles – 9,18)

     Response: The references have been corrected to indicate appropriate studies (references 6 and 17)

Round 2

Reviewer 3 Report

Thank you for the effort to improve the article. In my opinion, it could be now accepted for publishing.